# Modified Technique for Wirsung-Pancreatogastric Anastomosis after Pancreatoduodenectomy: A Single Center Experience and Systematic Review of the Literature

**DOI:** 10.3390/jcm10143064

**Published:** 2021-07-11

**Authors:** Cinzia Bizzoca, Salvatore Fedele, Anna Stella Lippolis, Fabrizio Aquilino, Marco Castellana, Maria Raffaella Basile, Giuseppe Lucarelli, Leonardo Vincenti

**Affiliations:** 1Department of General Surgery “Ospedaliera”, Polyclinic Hospital of Bari, Piazza G. Cesare 11, 70124 Bari, Italy; dr.leonardo.vincenti@gmail.com; 2General Surgery Unit, National Institute of Gastroenterology Saverio de Bellis, Research Hospital, via Turi 27, Castellana Grotte, 70013 Bari, Italy; salvatorefedele.md@gmail.com (S.F.); fabrizioaquilino@inwind.it (F.A.); 3Department of General Surgery, San Paolo Hospital, via Capo Scardicchio, 70123 Bari, Italy; lippolis2012@gmail.com (A.S.L.); bslraffy@alice.it (M.R.B.); 4Unit of Research Methodology and Data Sciences for Population Health, National Institute of Gastroenterology Saverio de Bellis, Research Hospital, via Turi 27, Castellana Grotte, 70013 Bari, Italy; mcastellana01@yahoo.it; 5Andrology and Kidney Transplantation Unit, Department of Emergency and Organ Transplantation-Urology, University of Bari, 70124 Bari, Italy; giuseppe.lucarelli@inwind.it

**Keywords:** wirsung-pancreatogastric anastomosis, pancreaticoduodenectomy, pancreatic surgery, pancreatic cancer

## Abstract

Background: The mortality rate following pancreaticoduodenectomy (PD) has been decreasing over the past few years; nonetheless, the morbidity rate remains elevated. The most common complications after PD are post-operative pancreatic fistula (POPF) and delayed gastric emptying (DGE) syndrome. The issue as to which is the best reconstruction method for the treatment of the pancreatic remnant after PD is still a matter of debate. The aim of this study was to retrospectively analyze the morbidity rate in 100 consecutive PD reconstructed with Wirsung-Pancreato-Gastro-Anastomosis (WPGA), performed by a single surgeon applying a personal modification of the pancreatic reconstruction technique. Methods: During an 8-year period (May 2012 to March 2020), 100 consecutive patients underwent PD reconstructed with WPGA. The series included 57 males and 43 females (M/F 1.32), with a mean age of 68 (range 41–86) years. The 90-day morbidity and mortality were retrospectively analyzed. Additionally, a systematic review was conducted, comparing our technique with the existing literature on the topic. Results: We observed eight cases of clinically relevant POPF (8%), three cases of “primary” DGE (3%) and four patients suffering “secondary” DGE. The surgical morbidity and mortality rate were 26% and 6%, respectively. The median hospital stay was 13.6 days. The systematic review of the literature confirmed the originality of our modified technique for Wirsung-Pancreato-Gastro-Anastomosis. Conclusions: Our modified double-layer WPGA is associated with a very low incidence of POPF and DGE. Also, the technique avoids the risk of acute hemorrhage of the pancreatic parenchyma.

## 1. Introduction

Pancreatoduodenectomy (PD) is the only chance of cure for patients diagnosed with a tumor involving the periampullary region. Nowadays, thanks to high quality imaging for pre-operative assessment, advances in surgical techniques and perioperative management, the mortality rate after PD has decreased to 3–5% in dedicated centers. Nonetheless, the morbidity rate after PD remains elevated, even in highly specialized centers (30–50%) [1,2,3]. The most common complications after PD are pancreatic fistula (POPF, 2–20%,), delayed gastric emptying (DGE, 19–57%), abdominal collections (9–10%) and post-pancreatectomy hemorrhage (PPH, 1–8%) [1,2,3].

In current state-of-the-art pancreatic surgery, the primary goal is to reduce major complications (i.e., pancreatic fistula and delayed gastric emptying), allowing both early recovery and early discharge of the patient.

The accepted surgical procedures for PD in cases of primary or secondary malignant conditions involving the periampullary region are pylorus-preserving PD (PPPD) (or Traverso–Longmire) and the classical Kausch–Whipple PD. Although there are several reconstruction techniques, each involves a different pancreatic anastomosis. It is possible to perform a pancreatojejunostomy (PJA) on a single loop, as described by Child in 1944, a PJA on Roux-en-Y, described by Machado, or a pancreatogastrostomy (PGA) [4,5,6,7,8,9]. In both the PJA and PGA techniques, some surgeons perform a duct-to-mucosa anastomosis (Wirsung-pancreatojejunostomy or Wirsung-pancreatogastrostomy) and/or use a ductal stent (external or internal stenting) [10,11,12,13,14,15,16,17,18,19] There is still no consensus as to the best reconstruction technique for the management of the pancreatic stump. In fact, several meta-analyses have focused on pancreatic anastomosis, but no agreement has been reached on which is the “gold standard technique” [20,21,22,23,24,25,26,27,28].

Standard PGA provides for invagination of the pancreatic stump in the gastric cavity, thus exposing the pancreatic remnant to hemorrhagic complications. The aim of this study was to retrospectively analyze the morbidity rate in 100 consecutive PD reconstructed with WPGA, performed by a single surgeon applying a personal modification of the pancreatic reconstruction technique. In addition, a systematic review of the literature is reported, comparing our technique with the existing literature on the topic.

## 2. Materials and Methods

During an 8-year period (May 2012 to March 2020) at the Department of General Surgery “Ospedaliera” at the Polyclinic Hospital in Bari (Italy), a single surgeon L.V. performed 212 pancreatic resections, 124 of which were PD. We retrospectively analyzed 100 consecutive PD reconstructed with WPGA over this period of time. The study was designed and conducted respecting the STROBE guidelines for observational studies.

The indication for surgery was upfront resectable primary or secondary malignancy (92 patients), 6 cases of Intraductal Papillary Mucinous Neoplasia (IPMN), and 2 of chronic “mass-forming” pancreatitis. The preoperative assessment included CT scan, neoplastic markers (CA 19.9 and CEA), MRI in selected patients, and evaluation of the surgical risk according to the ASA classification (Table 1). No ASA IV patient was excluded if judged fit enough to undergo major abdominal surgery by the anesthesiologists. Preoperative biliary drainage was not routinely placed in resectable patients if bilirubin was ≤10 g/dl and/or jaundice lasted for less than one week.

The primary endpoint was to analyze the incidence of POPF and DGE. According to the 2016 ISGPF classification, any measurable volume of drained fluid on or after POD 3, with amylase levels 3 times the upper normal limit (3 × ULN), was defined as POPF only in cases of clinical relevance. Instead, an increased amylase value in drainage fluid without clinical consequences was defined as a biochemical leak (BL) [29]. 

According to the ISGPS definition, delayed gastric emptying (DGE) was diagnosed in cases needing insertion of a nasogastric tube (NGT) after POD 3 or intolerance to solid oral intake after POD 7 [2]. Secondary endpoints included overall and surgical morbidity and mortality within 90 days. We also analyzed median operative time, intraoperative blood transfusions, length of hospital stay, reoperation and readmission rates.

### 2.1. Operative Technique

The operations were performed by the same surgeon, LV, and the reconstruction technique was always a personal, modified technique for WPGA in this series of patients. Compared to the standard PGA technique, the most important modifications adopted since 1998 were the use of an external catheter instead of an internal stent and the “double-layer” anastomosis, where only the gastric seromuscular layer is sectioned and the hole for the duct-to-mucosa anastomosis is created when the “armed” stent is passed through the posterior gastric wall. Additionally, the surgeon modified the type of sutures used for the anastomosis, adopting an absorbable monofilament instead of a non-absorbable material.

After pancreatic section, the pancreatic remnant was mobilized 3 cm toward the tail, with accurate dissection from the splenic vein and the surrounding tissues. Before performing the pancreatic anastomosis, the main pancreatic duct was explored to check its patency. The seromuscular incision, performed with a scalpel on the posterior gastric wall, was slightly larger than the greatest diameter of the pancreatic stump (Figure 1A). Suture between the anterior pancreatic surface and the gastric seromuscular layer was performed using interrupted a monofilament absorbable suture, knotted on the external side to avoid decubitus on the pancreatic parenchyma.

An external stent, which is a catheter of 6 to 10 Fr in diameter, almost as large as the duct, was introduced into the Wirsung prior to performing the duct-to-mucosa anastomosis. Then, this catheter was anchored to the Wirsung section surface with a single absorbable stitch. The external stent facilitates the performance of the anastomosis as it avoids occlusion of the duct by the sutures.

The duct-to-mucosa anastomosis was performed with interrupted monofilament absorbable suture. This anastomosis was begun at the anterior circumference, placing one or more stitches in each quadrant, depending on the main duct diameter (Figure 1B). The stent was passed through the posterior and anterior gastric wall thanks to its edge “armed” with a Redon nail. This trans-gastric catheter allows gastric drainage during the first postoperative days (since it reduces the pressure on the anastomosis) through its intragastric holes.

Finally, the gastric seromuscular layer was sutured to the posterior pancreatic surface. A double suture was performed, anastomosing the main pancreatic duct to the gastric mucosa and anchoring the pancreas to the gastric wall protected by the gastric mucosa itself. In this way, corruption of the pancreatic parenchyma by gastric acid secretions was avoided and intraluminal hemorrhage prevented. The catheter was secured with a Vicryl Rapid stitch to the anterior gastric seromuscular layer. It was successively exteriorized through the anterior abdominal wall under the left costal arch and fixed with an external stitch at the end of the operation (Figure 1C). Two drainage tubes were always positioned.

### 2.2. Postoperative Care

After surgery, the patients usually returned to the surgical ward. The nasogastric tube was removed on POD 1 and the patient started oral intake on POD 3. The external stent was closed as soon as possible (when the output was minimal) and removed after POD 10, during recovery or after discharge.

### 2.3. Systematic Literature Review

#### 2.3.1. Search Strategy

A five-step search strategy was conducted in Pubmed. Firstly, we searched for sentinel studies. Secondly, we identified keywords. Thirdly, the following complete search strategy was employed: ((pancreas[Title/Abstract] OR pancreatic[Title/Abstract]) AND (stomach[Title/Abstract] OR gastric[Title/Abstract]) AND anastomosis[Title/Abstract]) OR (pancreatogastric[Title/Abstract] OR pancreaticogastric [Title/Abstract]). Fourthly, studies on WPGA as the reconstruction technique after PD were selected. Studies on different procedures, such as PGA, were excluded. Lastly, the references of the included studies were searched to find additional relevant papers. The last search was performed on 14 July 2020. No language restriction was adopted. Four investigators (CB, FA, MC, SF) independently searched for papers, screened titles and abstracts of the retrieved articles, reviewed the full-texts, and selected articles for inclusion.

#### 2.3.2. Data Extraction

The following information was extracted independently by two investigators (FA, SF) in a piloted form: (1) general information on the study (author, year of publication, country, study type, follow-up period, inclusion criteria, number of patients); (2) details of the surgical technique; (3) complications, including the number of patients with POPF (B-C), DGE, and PPH; (4) global morbidity. For each selected article, the main paper and Appendix A were searched; if data were missing, authors were contacted via email. Data were cross-checked and any discrepancy was discussed.

## 3. Results

The series included 57 (57%) males and 43 (43%) females (M/F 1.32), the mean age was 68 (range 41–86) years, and median BMI was 25 (range 18.5–39). Some of the patients had previously been treated in outlying hospitals: 19 patients underwent biliary drainage as a temporary solution to the jaundice: 1 percutaneous transhepatic biliary drainage, 17 biliary endoprosthesis and 1 sphincterotomy without endoprosthesis positioning because of technical problems, which caused acute iatrogenic pancreatitis. One patient was referred to our center after “palliative” GEA had been performed because he was considered unresectable.

A Kausch–Whipple PD was performed in 55 cases, while 45 patients underwent a PPPD, according to the indications for PD and extension of the tumor. Standard lymphadenectomy was always associated, and involved the removal of lymphatic groups 5, 6, 8a, 12a, b, 13, 14v and 17, according to the Japanese classification [30]. Extended lymphadenectomy (removal of additional lymphatic stations such as 8p, 9, 12p, 14a, 16a2, 16b1) was necessary in 11 cases (11%) to allow complete histological staging because of intraoperative evidence of suspicious lymph nodes. In 16 patients (16%), vascular resection was necessary because of portal vein or porto-mesenteric carrefour infiltration, and 8 patients (8%) underwent “extended” procedures with “en bloc” resection of surrounding organs. 

Operative data are reported in Table 2: the pancreatic characteristics were classified according to the ISGPS as Type A (not-soft pancreatic texture, duct size > 3 mm), Type B (not-soft, ≤3 mm), Type C (soft pancreatic texture, duct size > 3 mm) or Type D (soft, ≤3 mm) [31]. Additionally, intraoperative patients’ characteristics were classified on the basis of the Fistula Risk Score, depending on pancreatic texture, pathology, Wirsung diameter and intraoperative blood loss (see Table 3) [32].

Eight patients (8%) developed a clinically relevant pancreatic fistula (5 grade B and 3 grade C) and 8 patients (8%) suffered a biochemical leak (Table 4). Grade B pancreatic fistulas were handled conservatively with enzymatic inhibitors and maintaining the drainage tubes until normalization of the amylase levels. One patient with a grade C fistula was reoperated and two patients died of POPF complications.

Three patients (3%) presented “primary” DGE, while in four cases (4%) we observed “secondary” DGE, related to abdominal collections or POPF. All cases were managed conservatively with prokinetics and/or repositioning of NGT for a few days. We observed only one case of prolonged DGE, which was solved with dietary re-education.

We observed 32 complications in 26 patients (26% morbidity, see Table 4). Seven patients (7%) developed “isolated” abdominal collections diagnosed by CT scan and were treated conservatively with antibiotics (in case of infection signs) and/or delayed drainage tubes removal. The amylase dosage on the drained fluid was negative for all of these patients.

Eight hemorrhagic complications were observed (8%): none of the patients experienced early-onset hemorrhage (in the first 24 h), whereas all cases are classified as late-onset PPH since they occurred more than 24 h postoperatively [3]. Three patients developed necrotic hemorrhagic pancreatitis (3%), in one this was associated with grade C POPF. Four cases of hemoperitoneum (4%) occurred. Two patients presented bleeding associated with grade B POPF: in one, this terminated spontaneously after treatment with percutaneous drainage, while the other was treated with embolization. Two patients were reoperated to achieve bleeding control. One patient (1%) suffered gastrointestinal bleeding due to gastric ulcer and was treated with high-dose pump inhibitors and repositioning of the NGT.

Additionally, one case of biliary fistula (1%) occurred but no cases of enteric fistula.

The surgical mortality was 6% (6 patients): three patients died of necrotic hemorrhagic pancreatitis, two patients due to grade C POPF complications and one patient, who underwent segmental venous resection, died of fatal vascular thrombosis.

The mean operative time was 224 (range 140–420) minutes and no intraoperative complications occurred. Nine patients (9%) received intraoperative blood transfusions. The reintervention rate was 3%: one patient was operated because of necrotic hemorrhagic pancreatitis complications and two patients because of hemoperitoneum. The median hospital stay was 13.6 days (range 7–55). Two patients were readmitted, one for hemoperitoneum caused by a pseudo-aneurysm of the hepatic artery and one for severe wound infection with wall dehiscence.

### 3.1. Systematic Review

#### 3.1.1. Flowchart

In total, 608 papers were found on PubMed and one additional article on a double-layer PGA was retrieved from the references of the examined studies [33]. Papers were analyzed for title and abstract; 503 records were excluded (outside the scope of the review (e.g., bariatric surgery, complications of PD, gastrectomy, gastroenteroanastomosis, surgeries other than PD, pancreas transplantation, pancreatic pseudocysts, surgical palliation), use of a technique other than WPGA, PGA without details on the surgical technique, PJA, POPF, endoscopic procedures, or animal studies). The remaining 106 papers were retrieved in full-text and 99 records were excluded because of the use of a technique other than WPGA (*n* = 34), no data of interest (*n* = 22), PGA without details on the surgical technique (*n* = 22), PJA (*n* = 19), overlapping data (*n* = 1), or patients undergoing surgical palliation (*n* = 1). Finally, seven articles were included in the systematic review, since they described a Wirsung-Pancreato-Gastro-Anastomosis (Figure 2) [14,16,17,19,33,34].

#### 3.1.2. Qualitative Analysis

The characteristics and technical details of the included studies are summarized in Table 5 [14,16,17,18,19,34]. The articles were published between 1984 and 2017 and had sample sizes ranging from 5 to 205 patients. Two studies were conducted in Japan, one in China, two in Egypt, one in Spain, and one in the United States of America. Participants were adult subjects who underwent pancreaticoduodenectomy (PD or PPPD) and reconstruction with “double-layer” WPGA, performed without anterior gastrostomy.

As regards the surgical technique, an internal stent was reported in five studies [14,16,17,19,34], while no stent was adopted by Telford et al. and El Nakeeb et al. [18,33]. Concerning surgical outcomes, rates of grade B-C POPF ranged from 0.7% to 15.5%, DGE from 4% to 21.6%, and global morbidity from 19% to 37.8%. PPH was reported in only three studies (see Table 5) [16,17,33]. The duration of follow-up was reported in only one study [33].

## 4. Discussion

Pancreatoduodenectomy is a challenging surgical procedure, even for skilled surgeons. Furthermore, an expert team of surgeons is necessary to adequately manage the postoperative course and complications, which may occur in a high percentage of patients (30–50%) [1,2,3]

Pancreatic reconstruction after PD can be performed by means of several techniques, but choosing between PGA and PJA is the main issue. In 1988, Icard [36] demonstrated the anatomical, technical and physiological advantages of PGA, a technique that had been described for the first time in 1946 by Waugh and Clagett [37]. The anatomical relationship between the stomach and pancreas creates perfect conditions for a tension-free anastomosis. Moreover, the good thickness and blood supply of gastric mucosa allows good vascularization and robustness of the suture, thus, better healing of the anastomosis [15,36,38]. The vertical position of the stomach prevents stagnation of gastric secretions, and consequently, tension on the anastomosis [14,27]. The distance between the biliary and pancreatic sutures reduces the risk of damage of the biliary anastomosis by pancreatic enzymes in cases of POPF [36,39]. Finally, the acid gastric environment and absence of enterokinases inhibit complete enzymatic activation and consequent damage of the anastomosis.

The most significant disadvantage of PGA is the increased risk of pancreatic hemorrhage, due to erosion of the pancreatic remnant by gastric secretions [8,9,10]. In order to prevent this life-threatening complication, the technique was modified by some authors by adding a Wirsung-mucosa anastomosis (from PGA to WPGA) [40].

The technique presented in this series is a double pancreatic suture. Since the gastric mucosa is interrupted only where the Wirsung-mucosa anastomosis is performed, the pancreatic stump is protected by the gastric mucosa against gastric acid secretions. In theory, the limit of this technique is the impossibility of performing it in cases of a markedly friable pancreatic parenchyma. Notably, when Wirsung cannulation cannot be achieved, WPGA on an external stent cannot be performed. Nonetheless, we succeeded in performing this anastomosis even in cases of a soft pancreatic parenchyma, after Wirsung duct cannulation. The catheter acted as a guide while performing the anastomosis, avoiding pancreatic duct occlusion by the sutures. Importantly, it works as internal perianastomotic drainage so it allows early removal of NGT. Pancreatogastric sutures were applied to anchor the pancreas to the stomach and reduce tension on the anastomosis.

Whichever technique is used for pancreatic reconstruction, POPF and DGE are the most common complications after PD, which mostly affect the postoperative course and the length of stay, increasing hospital costs [41].

Several meta-analyses of randomized controlled trials (RCTs) have focused on pancreatic anastomosis, although the issue as to which is the best reconstruction method for the treatment of the pancreatic remnant after PD is still a matter of debate [10,24,26,27,28,38,42]. Hallet et al. [38] showed that PGA is associated with a lower risk of fistula as compared to PJA, in both low- and high-risk patients (the risk reduction for POPF is 4% and 10%, respectively). Nonetheless, Guerrini et al. [10] demonstrated that PGA is associated with low fistula rates, but without reducing the overall rate of complications. Nonetheless, a recent meta-analysis on 15 RCT concluded that duct-to-mucosa pancreaticogastrostomy is associated with lower fistula rate, besides DGE syndrome, intrabdominal abscess and morbidity rate [28].

In our series, we observed 8 patients (8%) with a clinically relevant pancreatic fistula (grade B-C). In the literature, a grade B-C pancreatic fistula rate of 11 to 28.3% is reported [43,44,45,46,47]. In our series, one patient with POPF needed reoperation because of accidental pancreatic stent dislocation, and ensuing hemorrhagic shock and gastric perforation. Another patient, discharged on POD 15, was readmitted one week later because of hemorrhage due to a pseudo-aneurysm of the hepatic artery, treated with radiologic embolization. Two patients who underwent conservative treatment died because of POPF complications.

DGE syndrome is another important issue after pancreatic surgery, occurring in 19 to 61% of patients [2,48]; it often delays recovery and discharge of the patient. The ISGPS (International Study Group of Pancreatic Surgery) classified “primary DGE” as cases not related to abdominal complications and “secondary DGE” as cases associated with complications like fistula or abdominal collections [2,21]. Prevention of “secondary DGE” can be achieved by avoiding abdominal complications; instead, “primary DGE” could be related to the surgical technique itself. Klaiber et al. [49] showed, in a meta-analysis of 992 patients, that pylorus-resecting PD is not superior to pylorus-preserving PD for reducing DGE. This is in contrast with a Cochrane Review [50] that states that the Whipple operation significantly reduces the DGE incidence as compared with PPPD. A recent ISGPS review of 178 studies revealed average rates of DGE and clinically relevant DGE of 27.7% (range: 0–100%; median: 18.7%) and 14.3% (range: 1.8–58.2%; median: 13.6%), respectively [51]. It is remarkable that in our series only 7 patients (7%) developed DGE. These data include 3 patients (3%) with “primary DGE” and 4 patients (4%) with “secondary DGE”. In our series, secondary DGE was always related to POPF, and in some cases associated with abdominal collections. Despite these encouraging data, there is no evidence in the literature that PGA could prevent DGE syndrome [52].

Seven patients presented “isolated” abdominal collections (7%), not related to other abdominal complications (i.e., POPF). The diagnosis was made by CT scan, performed in four cases because of fever (4% abdominal abscesses) and in 3 cases for anemia (3% blood collections). All cases were handled conservatively, with no need for reoperation.

We observed 8 hemorrhagic complications (8%), all of them classified as late-onset, according to the ISGPS definition [3]. Four patients (4%) presented hemoperitoneum and two of them were reoperated: one on POD 2 to achieve bleeding control, another one on POD 11 because of hemorrhagic shock and gastric perforation, secondary to pancreatic catheter dislocation. Two patients with grade B POPF presented late-onset hemorrhage (>24 h postoperatively) [3]: one patient was managed conservatively, and treated with percutaneous drainage since no evidence of active bleeding was found at angio-CT scan; another patient was treated with radiologic embolization of the hepatic artery pseudo-aneurysm.

None of the patients experienced early-onset hemorrhage, defined as PPH raised within the first 24 h postoperatively [3]. Specifically, no case of acute pancreatic stump bleeding was observed.

Only one patient suffered a biliary leak, probably because in our reconstruction technique the distance between the pancreatic and biliary anastomoses protects the latter from pancreatic complications [36,39]. Six patients (6%) died of surgical complications (Table 4). One patient, who underwent segmental portal resection, developed post-operative venous thrombosis (PVT) and consequently died of acute hepatic failure. Three patients suffered acute necrotic hemorrhagic pancreatitis and died of the complications of this life-threatening condition. These data are remarkable, considering that post-operative pancreatitis could be considered a pancreatic stump complication, even with no evidence of a pancreatic fistula. Besides, hemorrhagic pancreatitis of the pancreatic stump is considered as a late-onset hemorrhagic complication in the ISGPS definition, associated with high mortality rate [3]. In fact, the ISGPS reports that post-operative pancreatic hemorrhage (PPH) accounts for 11 to 38% of overall mortality after pancreatic surgery [3]. Yekebas et al. [52] reported a prevalence of 5.7% of PPH in a series of more than 1500 patients with PJA reconstructions. Analysis of the cases of delayed PPH (after POD 6) revealed a mortality rate of 47%, while no patients with early PPH (up to POD 5) died. They concluded that the worst prognosis was associated with late-onset PPH, often related to POPF, causing erosions, pseudoaneurysms and other vascular irregularities, and consequently, life-threatening bleeding.

Roulin et al. [53] reviewed the incidence of delayed PPH (more than 24 h after pancreatic surgery) in 15 articles including 7400 patients. They found an overall incidence of 1.6 to 12.3% among different studies, and evinced that half of the cases were related to pancreatic leak. An overall mortality rate of 35% was reported by Roulin et al., which was caused by hemorrhagic or septic shock, disseminated intravascular coagulation and multiple organ failure [53]. A 43% mortality rate (3 of 7 patients) was observed in our series among patients with late-onset PPH. On the other hand, no cases of early-onset PPH occurred, therefore our modified technique eliminated the risk of acute pancreatogastric hemorrhage, which is the “Achille’s heel” of PGA [53]. Nonetheless, necrotic-hemorrhagic pancreatitis of the pancreatic stump remains a major clinical problem. Erosion of peripancreatic vessels is a recognized cause of delayed PPH [3,52,53]. This mechanism could explain the cases of acute necrotic-hemorrhagic pancreatitis we observed in our series, despite the absence of biochemical or clinically evident POPF. In fact, a “silent” disruption of the suture between the Wirsung duct and gastric mucosa could create the conditions for pancreatic damage due to gastric secretions. Comparing our results with the literature, whatever the technique used for pancreatic anastomosis, pancreatic surgery has not yet reached the goal of eliminating the risk of hemorrhagic complications after PD, and PPH remains the main cause of mortality (up to 50%) after pancreatic resection [52,54]. In any case, when comparing PGA with PJA, pancreatogastric anastomosis is associated with a higher incidence of PPH [22].

The reintervention rate was 3% (3 patients). One patient underwent abdominal toilet for necrotic-hemorrhagic pancreatitis complications but died despite reoperation. Two patients suffered abdominal bleeding and were reoperated for hemorrhagic shock. One of them presented with pancreatic stent dislocation two weeks after surgery.

The median hospital stay was 13.6 (7–55) days. Non-complicated patients were discharged after 7–10 days, thanks to postoperative management based on ERAS (enhanced recovery after surgery) protocols, which are strongly recommended after pancreatic surgery [15,16]. Furthermore, we observed a low rate of major complications (Clavien–Dindo grade III-IV, see Table 6), which usually affect the patient outcome, length of stay and hospital costs [41].

The limits of our study are represented by its retrospective nature and the small number of patients and complications observed. Also, this is the personal modification of a single experienced surgeon, who started with pancreaticoduodenectomy in 1998. Therefore, we cannot state that our technique for WPGA is widely reproducible or could reduce the rate of complications if adopted by other groups of surgeons.

### Comparison with Available Literature on WPGA

The characteristics and technical details of the Wirsung-Pancreato-Gastro-Anastomoses included in the review are described in Table 5, according to the ISGS classification and compared with our technique [35].

Telford and Mason [18] describe a WPGA without stenting. The pancreatic stent ensures the patency of the Wirsung duct when anastomosis is performed, thus avoiding its occlusion by the sutures. This is very important, especially when the duct to be sutured is small (<3 mm).

Takao and Shinchi et al. [19,34] reported their experience of the same surgical division. Compared with our technique, the main differences are the transfixing suture and the internal stent. In our experience, the use of the external pancreatic stent has the advantage of permitting early removal of NGT and resumption of oral intake. In fact, it allows external drainage of pancreatic and also gastric secretions, thanks to its gastric holes.

The techniques reported by the remaining authors (four of seven in our review) involve different pancreatic sutures (i.e., running sutures instead of single stitches)—see Table 5. Furthermore, none of them used external stents [14,16,17,33]. More importantly, we do not section the gastric mucosa, as all of the authors described. The hole for the duct-to-mucosa anastomosis is created when the “armed” stent is passed through the posterior gastric wall, where the mucosal integrity has to be preserved at the time of seromuscular section. This technical detail avoids redundancy of gastric mucosa and reduces the risk of discontinuity between the gastric mucosa and the Wirsung duct.

In the reviewed papers, we found that there was often a lack of information about the study inclusion and exclusion criteria (i.e., ASA score), Wirsung diameter (median, range), pancreatic parenchyma or other characteristics that could be related to surgical results. Furthermore, Shinchi et al. [19] did not use the ISGPS definition for POPF. Finally, the follow-up period was 90 days in our series, but was not specified in most of the reviewed articles, except for El Nakeeb et al. [33].

The difference in defining postoperative complications, as well as the lack of information about patient characteristics, exclusion criteria and follow-up period, could influence data collection, and therefore, a correct interpretation of postoperative outcomes. Therefore, we avoided comparing our results to other WPGA found in the literature. On the other hand, the aim of this review was to confirm the originality of our modified double-layer Wirsung-Pancreato-Gastro-Anastomosis.

## 5. Conclusions

Nowadays, pancreatic surgery offers satisfactory results in terms of mortality, but morbidity after PD remains elevated, whatever the technique used for treatment of the pancreatic stump. Despite all the guidelines and recommendations, currently there is no general consensus in the literature about the “gold standard” reconstruction technique to reduce the rate of POPF, DGE and surgical morbidity after PD. In this sense, the best technique is the one that reduces the risk of complications, especially pancreatic fistula and abdominal collections, and also the cases of “secondary DGE”.

Our modified double-layer WPGA, in which the gastric mucosa reduces the risk of pancreatic stump damage by the gastric acid secretions, is associated with a very low incidence of pancreatic fistula and DGE. Furthermore, this technique reduces the risk of acute hemorrhage of the pancreatic parenchyma as compared with PGA. Nonetheless, the goal to avoid life-threatening pancreatic stump complications, such as necrotic hemorrhagic pancreatitis, which is probably a consequence of “sub-acute” damage, has yet to be achieved.

Our results are limited by the retrospective nature of the study, besides the small number of complications observed. Further studies are needed to confirm the safety of our modified technique.

## Figures and Tables

**Figure 1 jcm-10-03064-f001:**
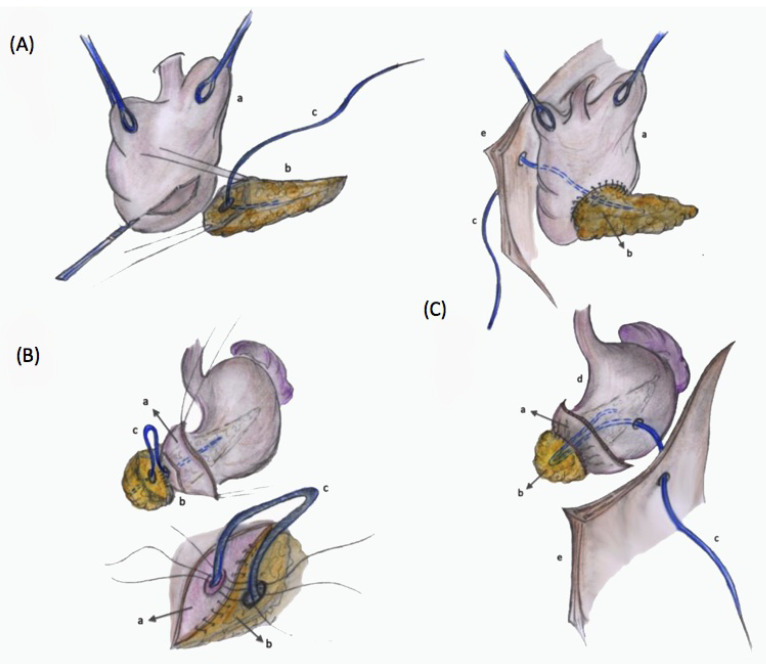
(**A**) The seromuscular incision on the posterior gastric wall (a) is slightly larger than the greatest diameter of the pancreatic stump (b). The catheter used as external stent (c) is introduced into the Wirsung duct to ensure its patency during anastomosis (**B**) The suture between the anterior pancreatic surface and the gastric seromuscular layer is performed. Then, the duct-to-mucosa anastomosis is performed, applying one or more stitches in each quadrant (**C**) After the anastomosis between the posterior gastric wall and the pancreatic stump is completed, the catheter is secured to the anterior gastric seromuscular layer (d) and exteriorized through the anterior abdominal wall (e).

**Figure 2 jcm-10-03064-f002:**
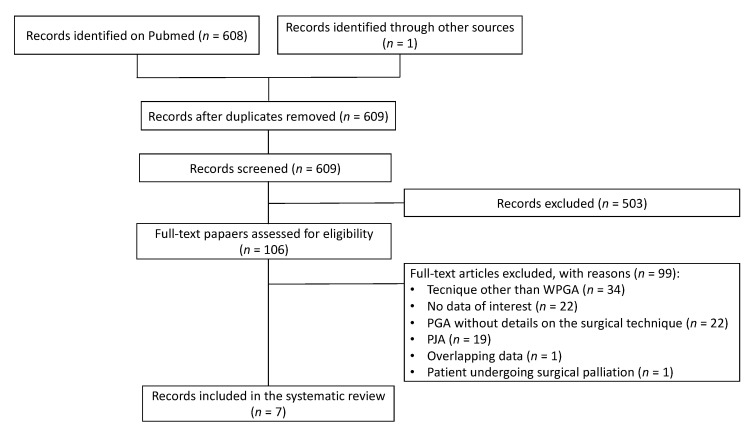
Flowchart of the systematic review. WPGA Wirsung-Pancreato-Gastro-Anastomosis; PGA Pancreatogastric Anastomosis; PJA Pancreatico-Jejunal Anastomosis.

**Table 1 jcm-10-03064-t001:** Pre-Operative Characteristics of the patients included in the study.

Characteristics	N. 100
Sex male female	57 43
Age (years)	68 (range 41–86)
Body mass index (kg/m^2^)	25 (range 18.5–39)
ASA SCORE I-II	58
III-IV	42
Comorbidity (Y/N)	
Hypertension (Y/N)	55/45
Diabetes mellitus (Y/N)	22/78
Other Y/N	59/41
Smoke (Y/N)	27/73
Preoperative biliary drainage Y/N	22/78
Preoperative total bilirubin (g/dl)	7
Previous palliative surgery	
Yes	2
ileocolic anastomosis (colon cancer)	1
palliative GEA (pancreatic cancer)	1
No	98

Y: yes, N: no.

**Table 2 jcm-10-03064-t002:** **Intraoperative****Data**.

Data	N. 100
Pancreatic Type [31]	
A	22
B	33
C	7
D	38
Type of PD	
Pylorus-preserving PD	45
Kausch–Whipple PD	55
Type of Lymphadenectomy	
Standard	89
Extended	11
Vascular Resection	
Yes	16
No	84
Extended surgery	
Yes	8
Right colectomy	5
Subtotal gastrectomy	1
Nephrectomy	2
No	92
Duodeno-jejunostomy/GEA	
Manual	100/9
Mechanical	0/91
Antecolic	98/92
Retrocolic	2/8
Intraoperative blood transfusion	
Yes	9
No	91
Operative time (min)	224
	(range 140–420)

Type A (not-soft pancreatic texture, duct size > 3 mm), Type B (not-soft, ≤3 mm), Type C (soft pancreatic texture, duct size > 3 mm) or Type D (soft, ≤3 mm). PD: pancreticoduodenectomy, GEA: gastro-enteral-anastomosis.

**Table 3 jcm-10-03064-t003:** Patients’ stratification according to the Fistula Risk Score [32].

Risk factors for Pancreatic Fistula	%
Gland texture	
Soft	45
Not-soft	55
Pathology	
Pancreatic adenocarcinoma or pancreatitis	61
Other (ampullary, duodenal, cystic, etc…)	39
Pancreatic duct diameter (mm)	
≥5 mm	5
4	28
3	43
2	24
<1	0
Intraoperative blood loss (mL)	
≤400 mL	90
401–700 mL	8
701–1000 mL	1
<1000 mL	1

**Table 4 jcm-10-03064-t004:** Surgical Morbidity: 32 complications occurred in 26 patients.

Surgical Morbidity	*n*°32 (26 pts)	%
Type of complication		
Pancreatic fistula *	8	(8%)
Grade B	5	
Grade C	3 (2 †)	
Delayed gastric emptying (primary/secondary)	7 (3/4)	(7%)
Grade A	2 (2/0)	
Grade B	1 (0/1)	
Grade C	4 (1/3)	
Abdominal collections	7	(7%)
Blood collection	4	
Abscess	3	
Haemorrhagic complications	8	(8%)
Early-onset [3]	0	
Late-onset [3]	8	
- Hemoperitoneum	4	
- Gastric ulcer	1	
- Necrotic haemorrhagic	3 (3 †)	
Pancreatitis		
Portal thrombosis	1 (†)	(1%)
Biliary fistula	1	(1%)
Enteric fistula	0	(0%)
Readmission	2	(2%)
- wound infection	1	
- haemorrhage	1	
Reoperation	3	(3%)
- Hemoperitoneum	2	
- Necrotic haemorrhagic pancreatitis	1	

Pts: patients; (†) Exitus; * According to the 2016 ISGPS definition [30]—Biochemical leak 8%.ù.

**Table 5 jcm-10-03064-t005:** Characteristics of the studies included in the systematic review [35].

First Author, Year	Country	Study Design	Study Period	Patients (*n*)	Procedure Details			Surgical Outcomes			
					PM	Stent	Suture/Details (1) Inner Layer (2) Outer Layer	POPF (B-C)	DGE	PPH	Global Morbidity
Telford, 1984 [18]	USA	Case series	1975–1977	5	-	S0	Interrupted (1) 4/0 silk (2) 4/0 PGA	-	-	-	-
Shinchi, 2006 [19]	Japan	RCS	1996–2004	103	PM2	S1	Transfixing (1) 2-0 silk (2) 4-0 absorbable	2%	6%	-	23%
Fernández-Cruz, 2008 [17]	Spain	RCT	2005–2007	53	-	S1	Gastric partition (1) continuous	4%	4%	2%	23%
Takao, 2012 [32]	Japan	RCS	-	205	PM2	S1	Transfixing (1) 2-0 silk (2) 4-0 absorbable	2%	7%	-	19%
El Nakeeb 2014 [33]	Egypt	RCT	2011–2013	45	-	S0	Interrupted or continuous (1) 3-0 silk (2) 5/0 vicryl	15.5%	20%	8.9%	37.8%
Osman, 2014 [14]	Egypt	RCS	2009–2013	37	PM3	S1	Purse string (1) 2/0 polypropylene + 3/0 polyglactin (2) 4/0 polypropylene	2.7%	21.6%	2.7%	29.7%
Lu, 2017 [16]	China	RCS	2010–2014	143	PM2	S1	Running (1) 4-0 polypropylene (2) 5-0/6-0 PDS	0.7%	5.6%	-	28.7%
Bizzoca et al.	Italy	Case series	2012–2020	100	PM3	S2	Interrupted (1) 3-0 PDS (2) 4-0/5-0 PDS	8%	7%	8%	26%

PM pancreatic mobilization PM1 < 1 cm, PM2 1–2 cm, PM3 > 2 cm; S0 no stent, S1 internal pancreatic stent, S2 external pancreatic stent; POPF (B-C), post-operative pancreatic fistula; DGE, delayed gastric emptying, PPH, post-pancreatectomy hemorrhage; RCS, retrospective cohort study; RCT, randomized controlled trial, -, not reported; PGA, polyglicolic acid; PDS, polidioxanone.

**Table 6 jcm-10-03064-t006:** Severe post-operative complications.

Clavien-Dindo Grade	N	Description	Treatment
Grade III	1	1 Hemoperitoneum	1 Percutaneous drainage
Grade IV	3	1 Hemoperitoneum 1 Pancreatic stent dislocation 1 Myocardial infarction	1 Reintervention 1 Reintervention 1 PTCA
Grade V	6	3 Necrotic Hemorrhagic Pancreatitis 2 grade C POPF 1 Portal vein thrombosis	2 Conservative tx 1 Reintervention 2 Conservative tx 1 Anticoagulant therapy

PTCA: Percutaneous Transluminal Coronary Angioplasty; POPF: Postoperative Pancreatic Fistula [29]. Tx: treatment.

## Data Availability

The data presented in this study are available on request to the corresponding author.

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
