# Peer review of "Modified Technique for Wirsung-Pancreatogastric Anastomosis after Pancreatoduodenectomy: A Single Center Experience and Systematic Review of the Literature"

_jcm, 2021, doi:10.3390/jcm10143064_

Round 1

Reviewer 1 Report

Modified technique for Wirsung-Pancreatogastric Anastomo-sis after Pancreatoduodenectomy: a single center experience and systematic review of the literature - Bizzoca et al. 2021

COMMENTS TO THE AUTHORS:

Thank you for giving me the chance to review this manuscript which addresses a clinically relevant topic. Postoperative pancreatic fistula and delayed gastric emptying are the main causes of postoperative morbidity in pancreatic surgery. In this study, Bizzoca et al. describe the use of a modified pancreatic anastomosis by a single surgeon.

Overall:

  • I highly recommend to check and use the STROBE guidelines for observational studies.

Abstract:

  • ‘’Our modified double-layer WPGA is associated with a very low incidence of POPF and DGE.’’ – please add statement on experience, single surgeon, learning curve etc. If I adapt this technique, would it immediately lower my POPF and DGE rate?
  • ‘’Also, the technique avoids the risk of acute hemor-rhage of the pancreatic parenchyma.’’ – please describe this data in the results section to support this statement

Introduction:

  • ‘’ only chance of care’’ or do you mean cure?
  • Please cite recent papers PMID: 33914473 33892952 28027816
  • ‘’ Standard PGA/WPGA provides for invagination of the pancreatic stump in the gastric cavity, thus exposing the pancreatic remnant to hemorrhagic complications’’ - it is kind of confusing to use ‘’invagination’’ for a ‘’duct-to-mucosa’’ anastomosis?

Methods:

  • Please check factors from PMID 19879614 to describe WPGA anastomosis
  • Add a explanation video of the anastomosis?
  • ‘’2.1. Operative Technique’’ – please specifically describe what the personal modification is for clarification
  • Please describe the surgeons experience with pancreatoduodenectomy and the modified WPGA anastomosis. Are the 100 patients the first 100 patients in whom this technique was used? Would be valuable to analyze the learning curve of this anastomosis (operation time, blood loss, morbidity etc).
  • Who did the data collection? What quality control measures were taken, especially for the main outcomes POPF and DGE? Please describe this.

Results:

  • Please explain ‘’intermediate’’ texture – somewhat unusual see PMID 33892952
  • Incidence of POPF and DGE is surprisingly low
  • ‘’Seven patients (7%) developed “isolated” abdominal collections diagnosed by CT scan and were treated conservatively, with antibiotics (in case of infection signs) and/or de-layed drainage tubes removal.’’ – No POPF?
  • Table 3 – please use PPH classification
  • Since the sample size is limited – I would suggest adding a detailed table with individual data of patients with POPF, DGE and PPH. How and when where they diagnosed, treated, outcomes etc
  • Table 4 – please comment for each study the differences with your anastomosis (or was it only the in/external use of stents?)

Discussion

  • Please add a specific paragraph describing the limitations
  • You mention table 5 – where is it?
  • Lines 114-132: I would suggest moving some of this to the methods section where you describe the anastomosis. To make it clear why it is different.

Reviewer 2 Report

The authors reported a single surgeon experience of 100 consecutive patients undergoing PD. Pancreatic remnant reconstruction was performed via a double-layer pancreato-gastro anastomosis with the peculiarity of an external stent positioning. The authors hypothesized that the advantage of fashioning this anastomosis can be twofold: reducing the erosion of the pancreatic stump and avoinding the occlusion of the Wirsung duct.

The primary endpoint of the study was the incidence of POPF and DGE.

In addition a systematic review of the literature was performed focusing on pancreatogastro anastomosis.

They concluded that their technique is associated with a very low incidence of POPF and DGE. Also, the technique avoids the risk of acute hemorrhage of the pancreatic parenchyma. 

The main topic of tis study is actual, considering the great attention for the mitigation strategies in pancreatic surgery and the authors need to be congratulated for they results. 

Hower some major concern resulted and need to be clarified:

  • why did the authors exclude patients other than upfront resectable?
  • in materials and methods the authors reported the pre-operative characteristics of the patients: these are results, not methods
  • page 9, raws 46-60: it would be more appropriate shift these data in the results paragraph
  • from the operative data seems that 61% of patients had a pancreatic texture other than soft; in a recently published paper from Schuh (A Simple Classification Of Pancreatic Duct Size and Texture Predicts Postoperative Pancreatic Fistula: A classification of the International Study Group of Pancreatic Surgery) soft pancreatic texture was the main risk factors for POPF and patients without soft pancreas had a clinical relevant POPF in less than 6% of cases. Considering that, 8% of CR-POPF reported by the authors in this cohort of patients does not seem so "very low"
  • please use a fisula risk score (i.e. A Prospectively Validated Clinical Risk Score Accurately Predicts Pancreatic Fistula after Pancreatoduodenectomy) in order to better stratify patients' population
  • The technical details of the inner layer (duct-to-mucosa) of the anastomosis are not clearly described
  • in a so wide literature a single centre, single surgeon, experience with such a little number of cases seems to have a very poor genereralizability, reproducibility and reliability. Please deepen the limits section

Reviewer 3 Report

Thanks to the authors for their interesting contribution to the field of pancreatic surgery and congratulations on the results obtained.

It was impossible to see the video clip sent as supplementary material because in second 41 the image is interrupted

Round 2

Reviewer 1 Report

Thanks for giving me the chance to review this paper.